

# The decline and fall of the mammalian stem

Neil Brocklehurst

Department of Earth Sciences, University of Cambridge, Cambridge, United Kingdom

## ABSTRACT

The mammalian crown originated during the Mesozoic and subsequently radiated into the substantial array of forms now extant. However, for about 100 million years before the crown's origin, a diverse array of stem mammalian lineages dominated terrestrial ecosystems. Several of these stem lineages overlapped temporally and geographically with the crown mammals during the Mesozoic, but by the end of the Cretaceous crown mammals make up the overwhelming majority of the fossil record. The progress of this transition between ecosystems dominated by stem mammals and those dominated by crown mammals is not entirely clear, in part due to a distinct separation of analyses and datasets. Analyses of macroevolutionary patterns tend to focus on either the Mammaliaformes or the non-mammalian cynodonts, with little overlap in the datasets, preventing direct comparison of the diversification trends. Here I analyse species richness and biogeography of Synapsida as a whole during the Mesozoic, allowing comparison of the patterns in the mammalian crown and stem within a single framework. The analysis reveals the decline of the stem mammals occurred in two discrete phases. The first phase occurred between the Triassic and Middle Jurassic, during which the stem mammals were more restricted in their geographic range than the crown mammals, although within localities their species richness remained at levels seen previously. The second phase was a decline in species richness, which occurred during the Lower Cretaceous. The results show the decline of stem mammals, including tritylodontids and several mammaliaform groups, was not tied to a specific event, nor a gradual decline, but was instead a multiphase transition.

## INTRODUCTION

The mammal lineage separated from the reptile-line amniotes during the Paleozoic between 315 and 330 million years ago (*e.g.*, *Crottini et al., 2012*; *Dos Reis et al., 2015*; *Laurin, Lapauze & Marjanović, 2018*; *Brocklehurst et al., 2021*). For the first 100 million years of their evolutionary history, a substantial diversity of stem mammal lineages dominated terrestrial ecosystems (*Rubidge & Sidor, 2001*; *Brocklehurst, Kammerer & Fröbisch, 2013*; *Brocklehurst et al., 2021*). These stem mammals include a range of clades and forms, some of which pass through the end-Permian and end-Triassic mass extinctions and persist in the fossil record throughout much of the Mesozoic (*Rubidge & Sidor, 2001*; *Tatarinov & Matchenko, 1999*). The identity and age of the youngest stem mammal is uncertain; they potentially survived into the earliest Cenozoic (*Goin et al., 2012*; *Huttenlocker et al.,*

Corresponding author
Neil Brocklehurst,
neilbrockpalaeo@gmail.com

*2018*; *Rougier, Martinelli & Forasiepi, 2021*). From this point until the present day, the descendants of a single common ancestor have consistently been the sole representatives of the mammal line, remaining today as the mammalian crown.

The mammalian crown, consisting today of monotremes, marsupials and placental mammals, originated during the Mesozoic. The precise timing of this origin is uncertain, due to the uncertain affinity of Triassic mammal-line fossils. *Von Huene (1940)* described a premolar from the Rhaetian (latest Triassic) of Germany, suggesting it had similarities to multituberculates (a now-extinct crown-mammal lineage). However, he did not include detailed comparison of the specimen with multituberculates, nor did he included a museum catalogue number, thus precluding further study of the specimen. The prevalence of specimens based on limited material assigned to crown mammal lineages persists into the Lower Jurassic, although many of these were more confidently assigned to species level (*e.g.*, *Prasad & Manhas, 2007*; *Montellano, Hopson & Clark, 2008*; *Parmar, Prasad & Kumar, 2013*). But it is still unclear how reliable an identification based on such material, dominated by isolated teeth, may be (*Sansom, Wills & Williams, 2017*; *Brocklehurst & Benevento, 2020*). Also known from the Rhaetian and Early Jurassic are haramiyidans, a lineage of mammals of uncertain affinity. Many phylogenetic analyses have found them to be within the mammalian crown, as an outgroup to the clade containing marsupials and placental mammals (*e.g.*, *Krause et al., 2014*; *Zheng et al., 2013*; *Bi et al., 2014*; *Zhou et al., 2019*). However, others have found haramiyidans to be stem mammals (*Rougier et al., 2007*; *Zhou et al., 2013*; *Luo et al., 2015a*; *Huttenlocker et al., 2018*; *Celik & Phillips, 2020*), while the analyses of *Krause et al. (2020)* and *King & Beck (2020)* found a polyphyletic Haramiyida, with Haramiyidae and Haramiyavidae (the families known from the Triassic) excluded from the mammalian crown. *Hoffmann et al. (2020)* recovered support for different hypotheses depending on the method of analysis used; parsimony and tip-dated Bayesian analyses found polyphyletic haramiyidans, while an undated Bayesian analysis found haramiyidans within crown mammals. The inclusion or exclusion of haramiyidans from the mammalian crown shifts the age of the mammalian crown node by between 25 and 40 million years (*Cifelli & Davis, 2013*; *Luo et al., 2015b*). Nevertheless, despite this uncertainty surrounding the origin of modern mammals, an Upper Triassic-Lower Jurassic origin of crown mammals is consistent with estimates derived from molecular clocks (*Meredith et al., 2011*; *Dos Reis et al., 2012*; *Dos Reis et al., 2015*) and tip-dating approaches (*Upham, Esselstyn & Jetz, 2019*).

The mammalian stem includes several lineages overlapping in time with the mammalian crown, including docodonts, tritylodontids, and morganucodontids (for summary see *Grossnickle, Smith & Wilson, 2019*). As with the earliest crown mammal, the youngest stem mammal is uncertain due to fragmentary material and uncertain phylogeny. If haramiyidans, or at least their youngest representatives, are crown mammals, then the youngest well-supported stem mammals are Lower Cretaceous tritylodontids and docodonts (*Tatarinov & Maschenko, 1999*; *Lopatin & Agadjanian, 2007*; *Leshchinskiy et al., 2000*; *Lopatin et al., 2009*). A single Upper Cretaceous non-mammalian cynodont has been recorded from the Cenomanian of Australia (*Musser et al., 2009*). The specimen includes a femur and tooth referred to the Probainognathia. Unfortunately, the only record of this

cynodont is a published abstract giving no information of the morphology of the specimen, and thus far no full description has been published, so it is difficult to assess the reliability of this assignment. Moreover, the abstract does not provide a museum repository number so the assignment cannot even be verified by direct examination of the specimen. Therefore, although this specimen has been cited in discussions of a relict fauna isolated in Australia during the Cretaceous (*Leahey & Salisbury, 2013*), it cannot be seriously considered in discussion of the youngest non-mammalian synapsids. Another supposed non-mammalian cynodont has been described from as late as the Paleocene: *Chronoperates paradoxus* (*Fox, Youzwyshyn & Krause, 1992*). However, subsequent discussion of this specimen has not widely accepted this assignment, with doubt cast on the presence of the postdentary bones, and the dental morphology suggested to differ from those of non-mammalian cynodonts (*Sues, 1992*; *McKenna & Bell, 1997*; *Meng et al., 2003*). It should also be noted that, if Haramiyida are stem, rather than crown mammals (as discussed above), then at least one mammalian stem lineage would be known to have survived through to the end of the Mesozoic and into the earliest Cenozoic (*Huttenlocker et al., 2018*), with gondwanatherian mammals (found by these authors within haramiyidans) known from the Eocene of South America (*Goin et al., 2012*; *Rougier, Martinelli & Forasiepi, 2021*).

Before the end of the Mesozoic, the mammal-line fossil record had transitioned from a set of ecosystems dominated by stem mammals to one where most or all specimens belong to the mammalian crown (*Grossnickle, Smith & Wilson, 2019*). A series of discrete radiations have been identified that set the seeds for the present diversification of modern mammals: the Jurassic radiation of theriimorphs (*e.g.*, *Luo, 2007*; *Close et al., 2015*), the mid Cretaceous radiation of therian mammals coinciding with the diversification of angiosperms (*e.g.*, *Grossnickle & Polly, 2013*; *Benson et al., 2013*), a late Cetaceous therian diversification (*Wilson et al., 2012*; *Newham et al., 2014*; *Grossnickle & Newham, 2016*), and the radiation of eutherian mammals following the end Cretaceous mass extinction (*e.g.*, *Slater, 2013*; *Wilson, 2013*; *O'Leary et al., 2013*; *Halliday, Upchurch & Goswami, 2016*; *Brocklehurst et al., 2021*; *Benevento, Benson & Friedman, 2019*; *Benevento et al., 2023*). Analyses of species richness and morphological diversity during the Mesozoic have also documented the decline of stem mammal lineages (*e.g.*, *Ruta et al., 2013a*; *Lukic-Walther et al., 2019*; *Varnham, Mannion & Kammerer, 2021*). However, such analyses have examined the modern mammals and the individual lineages of stem mammmals in isolation of each other. With the exception of *Hellert et al. (2023)*'s analysis of diet and body size across the major synapsid lineages, there has been no analysis documenting the diversification, either morphological or species richness, of Synapsida as a whole throughout the Mesozoic, rendering the transition between stem-dominated and crown-dominate ecosystems unclear. In particular, the decline to extinction of the different stem mammal lineages has been analysed in isolation (*e.g.*, *Ruta et al., 2013a*; *Ruta et al., 2013b*; *Lukic-Walther et al., 2019*) with more focus placed on the pattern of modern mammal radiations. Unlike, for example, birds, where the transition to crown-dominated ecosystems may be easily linked to the faunal turnover at the end-Cretaceous mass extinction (*Longrich, Tokaryk & Field, 2011*; *Prum et al., 2015*), there is no large-scale event that may be easily tied to the extinction of stem mammals. The lack of an analysis of crown and stem mammal
diversification patterns under a single framework prevents direct comparison of the two and prevents demonstration of either a gradual replacement of the stem lineages by ever more crownward taxa, a rapid turnover under a single event, or a multiphase replacement process.

I analyse synapsid species richness and distribution and their changes throughout the Mesozoic, studying both crown and stem mammals under a single analytical framework in order to identify the progress of the decline of the mammalian stem and the transition to crown dominated ecosystems. I compare their species richness within areas of endemicity (bioregions) and present a novel approach for comparing geographic range size under incomplete sampling.

## MATERIALS & METHODS

### Dataset

The dataset of Synapsida from the Triassic until the end of the Cretaceous was downloaded from the Paleobiology Database (PBDB; https://paleobiodb.org/) on the 13th April 2023. The dataset includes all body fossils occurrences, excluding trace fossils (Data S1). The occurrences were divided between crown and stem mammals using the Paleobiology Database's higher taxonomy, which classes crown mammals as the order "Mammalia", and stem mammals as "Osteichthyes". The assignments to these higher taxa were checked to ensure they were accurate and up to date, as were the details of the occurrences and specimen identities. The decision on what clades represent stem or crown mammals were derived primarily from *Zhou et al. (2019)*, *Huttenlocker et al. (2018)* and *King & Beck (2020)*, which include comprehensive phylogenetic analyses that broadly sampled the key clades and had reasonably high support for the key nodes. All changes made to the dataset and their justification are detailed in Data S2. These include: deletion of two occurrences that are not considered reliable occurrences of Mesozoic synapsids; reassignment of six occurrences from crown to stem mammals; deletion of three occurrences where specimens unnamed in their original description were later assigned species names, but duplicate occurrences were created in the PBDB; deletion of 12 occurrences that represent trace records but were listed in in the download as body fossil records. The stage ages were updated to reflect the most recent timescale of the International Commission of Stratigraphy.

From this dataset, three test datasets were created, with different assignments for Haramiyida as discussed above. The first assigned Haramiyida (including Gondwanatheria, following *Huttenlocker et al. (2018)*) to the mammalian crown (following *Bi et al. (2014)* and *Zhou et al. (2019)*); the second to the mammalian stem (following *Luo et al. (2015a)*, *Luo et al. (2015b)* and *Huttenlocker et al. (2018)*); the third treated Haramiyidae and Haramiyaviidae as stem mammals and the remaining Haramiyida as crown mammals (following *Krause et al. (2020)*, *King & Beck (2020)*). All analyses described below were carried out on all three datasets to compare the results (Data S3–S5). The three datasets were passed through the *check_taxonomy()* function of the R package fossilbrush (*Flannery-Sutherland et al., 2022*) in R v 4.1.3 (*R Core Team, 2022*).

## Analysis of local diversity

The species richness of stem and crown mammals was assessed at the local, rather than global scale. Assessments of local diversity have the advantage that they represent comparisons of species that could have interacted in life (*Bambach, 1977*; *Close et al., 2019*), as well as mitigating the issue of spatial sampling heterogeneity that affects global diversity estimates, where a wider geographic spread can sample species from a broader range of bioregions and thus increase the diversity estimate (*Benson et al., 2016*; *Benson et al., 2021*; *Close et al., 2017*; *Close et al., 2020*). The unit of spatial sampling used in this study is the bioregion, a continuous geographic area containing a distinct assemblage of species (*Sclater, 1858*; *Wallace, 1876*). Using such areas of endemicity ensures that the units of sampling are biologically meaningful, representing assemblages of cohabiting species separated from other such assemblages (*Morrone, 1994*; *Linder, 2001*; *Oliveira, Brescoit & Santos, 2015*; *Elder et al., 2017*; *Ferrari, 2017*; *Brocklehurst & Fröbisch, 2018*).

The bioregions were defined according to the method described by *Brocklehurst & Fröbisch (2018)*: within each time bin, the collections defined in the PBDB were grouped in two hierarchical cluster analyses: the first by geographic distance and the second by taxonomic distance. Clusters found in both cluster analyses represent continuous geographic areas with a distinct set of taxa. The geographic distances between collections were based on the palaeo-coordinates provided in the PBDB download, and were calculated using the *lets.distmat()* function in the letsR package (*Vilela & Villalobos (2015)*; all analytical code is presented in Data S6). The taxonomic distances were calculated using the modified Forbes metric (*Alroy, 2015*),  using code provided in *Brocklehurst, Day & Fröbisch (2018)*. The bioregions in each time bin were calculated using a modified version of the code provided in *Brocklehurst & Fröbisch (2018)*. The bioregions were grouped at a cluster node height of 100km *i.e.,* collections with consistent faunas within 100km of each other were grouped into bioregions. This ensures that the bioregions represent local areas, and that the spatial scale of the analysis is maintained between time bins (*Brocklehurst & Fröbisch, 2018*).

Within each bioregion, occurrence counts of each synapsid species were extracted. Higher taxa were also included, provided no occurrences of subtaxa were found within the region, *e.g.*, if an occurrence is not named to species level, but is assigned to the genus *Thomasia*, and no occurrences of *Thomasia* named to species level are recorded, then the genus-level occurrence of *Thomasia* is included as a separate taxon. If, however, there is one occurrence of *Thomasia hahni*, and another assigned only to the genus *Thomasia*, then the genus-level occurrence is discarded, as it is not clear if it represents another occurrence of *T. hahni* or a different species of *Thomasia*.

Species richness of crown and stem mammals within each bioregion was estimated using shareholder quorum subsampling (*Alroy, 2010*; *Chao & Jost, 2012*), where occurrences are subsampled to a fixed level of coverage to produce more accurate estimates of relative richness. This was carried out using the *estimateD()* function in the R package iNEXT (*Hsieh, Ma & Chao, 2016*). A coverage quorum of 0.9 was applied, as using coverage levels lower than this produces imprecise results (*Close et al., 2018*; *Brocklehurst & Fröbisch,*

*2018*). The mean of the subsampled richness of stem and crown mammals in all bioregions in each time bin was used to compare local diversity trends.

The above means of comparing diversity trends between stem and crown mammals includes all bioregions in each time bin with sufficient sampling, whether they contain both or just one of the two groups. Thus, the observed trends will be influenced both by the geographic ranges of the two groups as well as the overall spatial sampling (more discussion of this issue below). An alternative, direct comparison of the relative diversities of stem and crown mammals within each bioregion was carried out. Within each time bin, bioregions containing both stem and crown mammal occurrences were identified and the ratio of crown to stem mammals calculated.

## Geographic range and spatial sampling

A group's decline in geographic range may not necessarily be concurrent with their decline in species richness; they may remain abundant and diverse in local areas, but have a more restricted geographic range, or vice versa. Therefore, it is necessary to examine the geographic spread of the localities in which stem mammals are found to obtain a complete picture of their decline relative to crown mammals. However, assessment of geographic range is hampered by heterogeneity in geographic sampling, with a wider spread of sampling in certain time intervals relative to others (*Benson et al., 2016*; *Benson et al., 2021*; *Close et al., 2017*; *Close et al., 2019*). It is therefore necessary, when comparing the trends in geographic range of crown and stem mammals, to account for the geographic spread of sampling of synapsids.

The geographic spread of sampling of synapsids was assessed by calculating the minimum spanning tree (MST) length between each collection in each time bin, as recommended by *Close et al. (2017)*. The minimum spanning tree represents the shortest branching network that connects all points in space. By summing the length of each segment connecting two points, one may obtain an estimate of the total spatial distance represented by the sample, incorporating a combination of spatial sampling signals (*Close et al., 2017*). The minimum spanning tree was calculated using the *mst()* function in the R package ape (*Paradis, Claude & Strimmer, 2004*). The minimum spanning tree length was also calculated for localities containing crown and stem mammals. These latter values were divided by the total synapsid MST length, to indicate the proportion of the geographic sampling of synapsids that contained either crown or stem mammals. This allows estimates of their ranges in different time bins to be directly comparable, as they are normalised by the total geographic spread of sampling of synapsids, rather than comparing observed range sizes that would be subject to inconsistent patterns of spatial sampling between time bins.

## RESULTS

### Local diversity

The local diversity estimates of stem and crown mammals during the Triassic and early stages of the Jurassic are broadly consistent across all three datasets (Fig. 1). Following a peak in the Lower Triassic, stem mammals experience an interval of consistent mean diversity, with similar ranges of values within each bioregion, throughout the Triassic and initial
stages of the Lower Jurassic. There is a peak in mean diversity of stem synapsids observed in the Rhaetian, but this is largely driven by a single exceptional locality: Habay-la-Vieille II in Belgium, of the Sables de Mortinsart Formation (*Godefroit, 1999*). This locality contains numerous synapsid teeth obtained by sieving; it may be that the collection method enables the collection of larger number of species than macro-vertebrate assemblages, or that the naming of numerous species based on limited material has resulted in an over-split taxonomy (*Lukic-Walther et al., 2019*). Excepting this locality, other bioregions have diversity estimates within the range observed in other Triassic stages. The Rhaetian is also the only Triassic stage in which the sampling of crown mammals is sufficient to make a diversity estimate in all three datasets, and in all three it is found to be lower than that of stem synapsids. Rhaetian localities containing both stem and crown mammals are primarily dominated by stem mammals, although if haramiyidans are considered crown synapsids a minority of localities contain greater species richness of crown mammals (Fig. 2). During the initial stages of the Lower Jurassic, it is only when all haramiyidans are included within crown that crown mammals have sufficient sampling to calculate diversity, which is lower than that estimated for the Rhaetian, and remains lower than that of stem mammals (Fig. 1A).

Through the Middle Jurassic the ranges of local diversity values observed in stem and crown mammals are consistent, with mean estimates extremely similar (Fig. 1). If haramiyidans are considered crown mammals or polyphyletic, mean diversity of crown mammals is found to be higher during the Middle Jurassic, but the difference is not substantial. In localities containing both stem and crown synapsids, the proportion of species represented by crown and stem mammals are on average roughly equal (Fig. 2). The mean diversity of both decreases briefly across the transition to the Upper Jurassic, but that of crown mammals rapidly increases and is substantially higher than that of stem mammals for the rest of the Upper Jurassic (Fig. 1). In all three datasets, localities containing both crown and stem mammals have on average nearly 80% of species representing crown mammals (Fig. 2).

All three datasets suggest that mean crown mammal diversity falls below that of stem mammals during the early stages of the Lower Cretaceous (Fig. 1). It should be noted, however, that for this interval stem mammals are only known from one locality of uncertain age (see Discussion), so it is unclear to what extent this represents a true signal, or whether the higher mean diversity of stem mammals is a quirk of a single community. Within this locality, between 20 and 40% of species are crown mammals, depending on whether haramiyidans are considered crown or stem (Fig. 2). The reduced diversity of crown mammals is observed in multiple localities, with both the mean and maximum local diversity lower than at any point in the Jurassic (Fig. 2). The mean diversity of crown mammals recovers during the Aptian and Albian, remaining at a similar level for the rest of the Cretaceous. With the exception of haramiyidans and the uncertain Australian cynodont (see Introduction) no stem mammals are known from the Upper Cretaceous; if haramiyidans are considered stem synapsids, then their mean diversity in the latest Cretaceous is considerably lower than that of crown synapsids, and lower than at any other point in the Mesozoic (Fig. 1B). However, if haramiyidans are considered stem

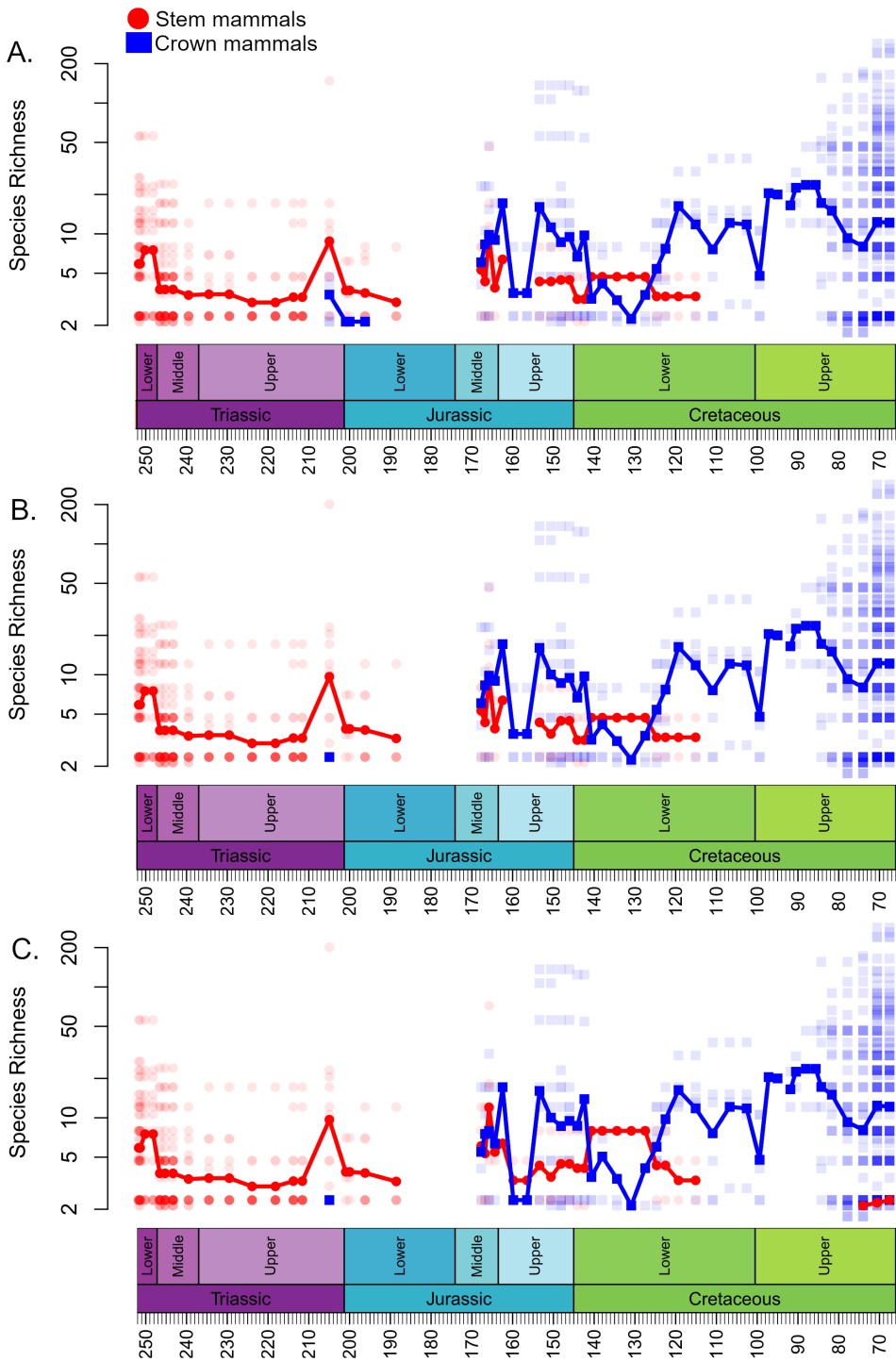

**Figure 1** **Species richness (subsampled) of Mesozoic crown and stem mammals through time.** Each translucent point represents the diversity within a bioregion. The solid line represents the mean diversity in each time bin. (A) Haramyida are included in crown mammals. (B) Haramyida are included in stem mammals. (C) Haramyida are polyphyletic, with Haramyidae and Haramyavidae included in stem mammals and others included in crown mammals.

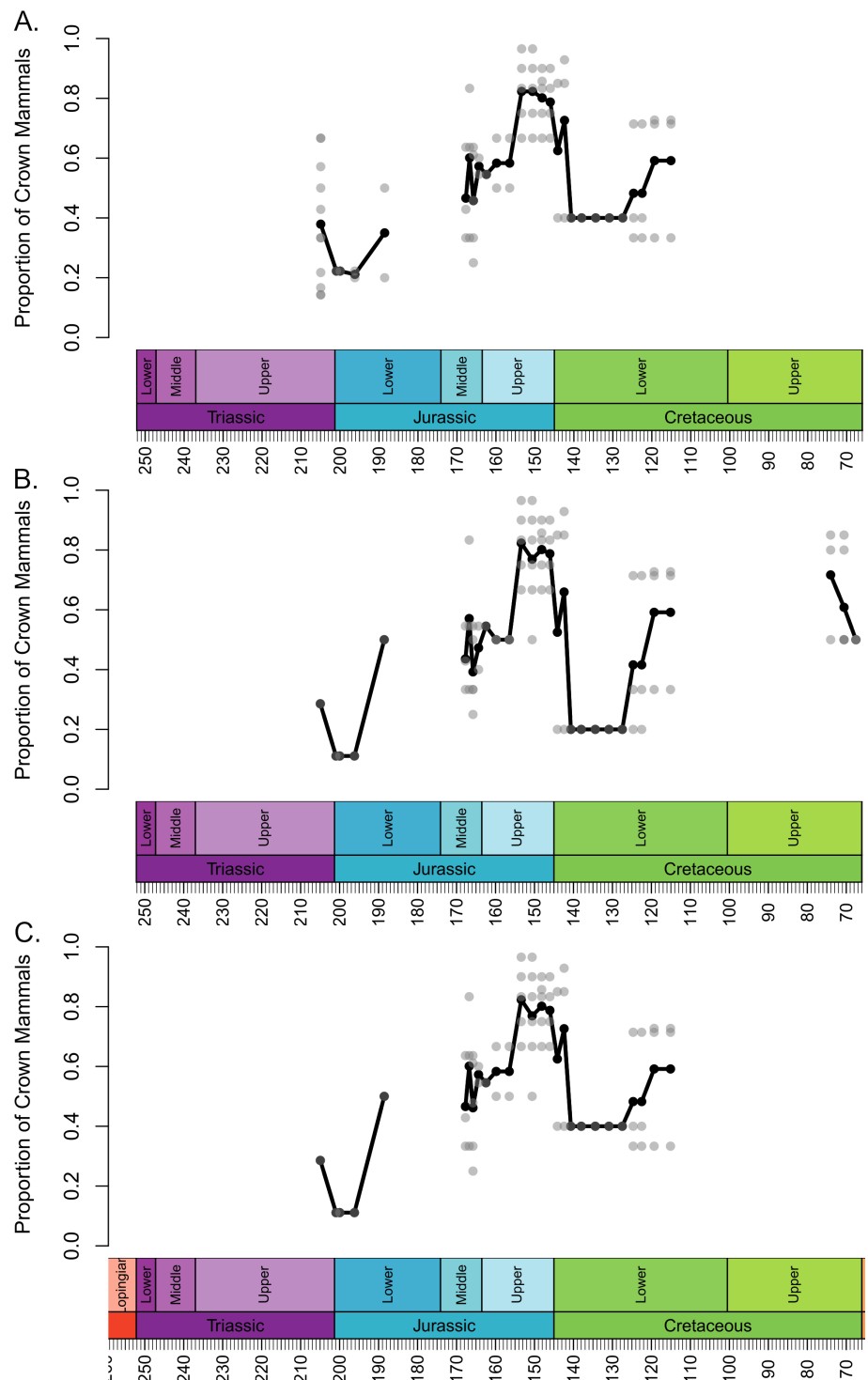

**Figure 2 Proportion of species representing crown mammals within bioregions containing representatives of both the crown and stem.** Each translucent point represents the proportion within a bioregion. The solid line represents the mean proportion in each time bin. (A) Haramyida are included in crown mammals. (B) Haramyida are included in stem mammals. (C) Haramyida are polyphyletic, with Haramyidae and Haramyavidae included in stem mammals and others included in crown mammals.

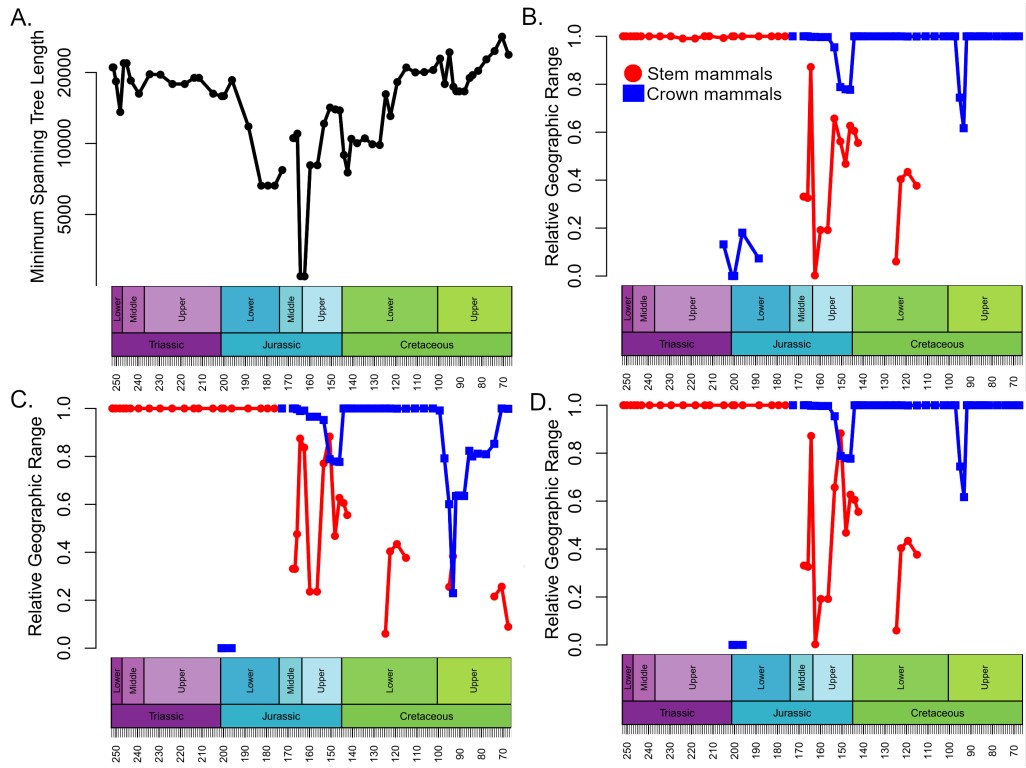

**Figure 3** Spatial sampling and geographic range of crown and stem mammals. (A) Minimum spanning tree length between all collections containing synapsid fossils within each time bin (a proxy for the spatial extent of sampling). (B–D) The proportion of the minimum spanning tree length representing collections containing crown mammals (blue) and stem mammals (red). (B) Haramyida are included in crown mammals. (C) Haramyida are included in stem mammals. (D) Haramyida are polyphyletic, with Haramyidae and Haramyavidae included in stem mammals and others included in crown mammals.

mammals, then in localities containing representatives of both the crown and the stem, crown mammals make up between 60 and 80% of species (Fig. 2B).

## Geographic range

The geographic ranges of crown and stem mammals, normalised relative to the total spatial sampling of synapsids, show broadly consistent patterns through time in all three datasets (Fig. 3). Through the Triassic and Lower Jurassic, the stem mammals are found across almost all of the geographic range in which synapsids are found. Following their appearance in the Rhaetian, the crown mammals occupy only a small portion of this range. Between the Lower and Middle Jurassic there is an abrupt increase in the geographic range of crown mammals, which occupy nearly the total range of synapsid sampling. The range of stem mammals decreases and is more restricted than that of crown mammals for most of the Middle and Upper Jurassic. The range of stem mammals further contracts during the Lower Cretaceous. Even when haramiyidans are considered stem synapsids, by the Upper Cretaceous their observed range represents only about 20% of the total geographic sampling of synapsids (Fig. 3B).

## DISCUSSION

The analyses do not tie the replacement of stem mammals with the crown lineages to a single event. Instead, two distinct phases may be identified. The first phase occurred during the Jurassic and represents a restriction on the geographic range of the stem mammals relative to the crown, but with little decrease in their local species richness. It is unclear exactly when this happened; across the Triassic/Jurassic boundary, stem mammals are found across almost all the geographic sampling range of synapsids (Fig. 4A), while crown mammals have a very limited range. For much of the Lower Jurassic, there is not sufficient sampling to assess the relative ranges of the two groups, but by the Middle Jurassic, the geographic range of crown mammals has substantially increased and is greater than that of the stem mammals in all three datasets (Figs. 3 and 4B). This remains the case for most of the remainder of the Mesozoic. By the Middle Jurassic, crown mammals have spread across both Laurasia and Gondwana (Fig. 4B). At this same time, however, stem mammals become increasingly rare in the fossil record across the southern continents of Gondwana. In fact, if haramiyidans are assumed to belong to the mammalian crown, then the only stem mammals known from Gondwanan localities from the Middle Jurassic until the end of the Cretaceous is the putative Probainognathan cynodont from Australia discussed in the introduction.

Despite this geographic restriction, the species richness of stem mammals within the regions in which they are found does not show any substantial decrease. Both the mean and maximum richness of stem mammals within localities as late as the Upper Jurassic remain at levels observed through much of the Triassic (Fig. 1). Even as crown mammals were becoming more species rich, both globally and within localities which they share with stem mammals, there is little appreciable change in the diversity of the stem. This result implies that the geographic restriction of the stem mammals is not due to competitive replacement by the crown mammals; if it were, one would expect the median richness of stem mammals to decrease in the Middle and Upper Jurassic localities that they share with crown synapsids. It appears instead that the rapid radiation and spread of crown mammals during the Middle Jurassic was independent of the decline of stem mammals. Either the stem mammals were eliminated from localities by other factors *e.g.*, shifting climate, competition with reptile groups radiating at the time, and the crown mammals dispersed into these regions subsequently, or the crown mammals entered regions already occupied by stem mammals, but occupied different ecological niches. Stem mammal lineages such as tritylodontids and docodonts include medium-sized herbivores (*Kalthoff et al., 2019*), aquatic piscivores (*Ji et al., 2006*), specialised burrowers (*Luo et al., 2015b*), and arboreal insectivores (*Meng et al., 2015*), niches which crown mammals would not occupy until the Cretaceous (with the exception of Haramiyida, of uncertain stem/crown affinity, which includes herbivorous and arboreal forms (*Meng et al., 2014*)).

The precise timing of the appearance and spread of crown mammals is not only obscured by the geographically patchy record of the Lower Jurassic, but also by a size-based bias in preservation. Prior to the Rhaetian (latest Triassic), the Triassic record is dominated by large-bodied synapsids, with small-bodied taxa being rare and fragmentary (*Lukic-Walther*
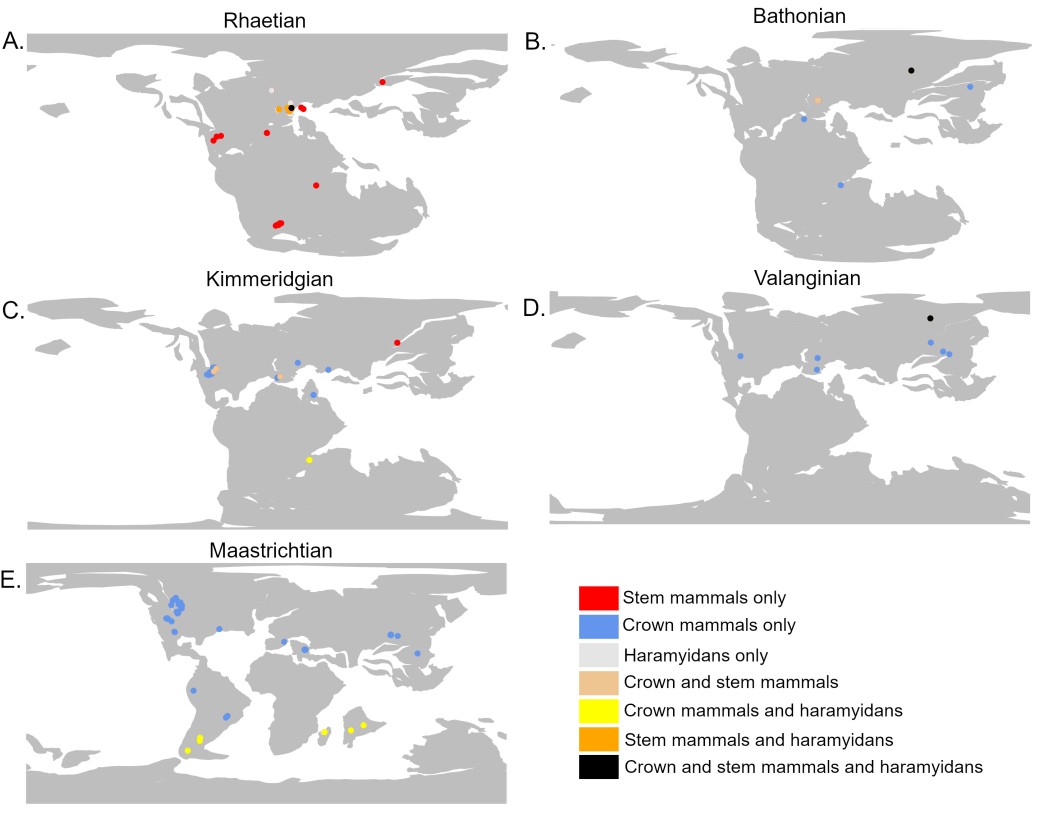

**Figure 4** **The geographic location of mammal fossils on palaeocontinental reconstructions.** Created using the R package Chronosphere (*Kocsis & Raja, 2023*). (A) Rhaetian (Late Triassic); (B) Bathonian (Early Jurassic); (C) Kimmeridgian (Late Jurassic); (D) Valanginian (Early Cretaceous); (E) Maastrichtian (Late Cretaceous).

*et al., 2019*). The sudden peak in Rhaetian synapsid diversity (Fig. 4), is likely due to increased sampling of micro-vertebratesin the latest Triassic, driven by study of mammals and lepidosaurs (*e.g.*, *Sigogneau-Russell & Hahn, 1994*; *Heckert, 2004*; *Van den Berg et al., 2012*). Such sites are not absent from earlier in the Triassic but have been less frequently exploited (*Gaetano et al., 2012*; *Lukic-Walther et al., 2019*). Thus, there may be a diversity of small-bodied synapsids unsampled from earlier in the Triassic.

The second phase in the decline of stem mammals, where the diversity of stem mammals within each locality falls consistently below that of crown mammals, is difficult to date reliably. The incomplete spatial sampling during the Upper Jurassic and Lower Cretaceous makes it difficult to assess to what extent the signals observed are local exceptions as opposed to true global signals. The Upper Jurassic peak in crown mammal species richness relative to that of stem mammals does coincide with morphological diversification of eutherian mammals (*Close et al., 2015*; *Grossnickle, Smith & Wilson, 2019*) but may in part be driven by a few exceptional localities, such as the Guimarota largerstätte (*Martin, 2001*), and the Morrison Formation localities at Como Bluff (*Prothero, 1981*; *Carrano & Velez-Juarbe, 2006*) and Dinosaur National Monument (*Engelmann, 1991*). The apparent

dominance of crown mammals at this time may be a local signal representing patterns in these localities/regions, especially since areas outside North America and Europe are particularly poorly sampled in this time interval. This peak may also be influenced further uncertainty in the phylogenetic relationships of mammals, in particular the relationships of Eutriconodonta, a diverse lineage of mammals included within the crown in this study (following *Zhou et al., 2019*; *Huttenlocker et al., 2018*; *King & Beck, 2020*), but which some have suggested they are a potentially paraphyletic series of outgroups to the crown (*e.g.,* *Rougier, Apesteguía & Gaetano, 2011*; *Krause et al., 2020*; *Celik & Phillips, 2020*), mostly based on postcranial characters. Eutriconodonta have been poorly represented in most phylogenetic analyses, but represent a diverse lineage in the Mesozoic, particularly during the mid-late Jurassic (*Butler & Sigogneau-Russell, 2016*). It was decided to include Eutriconodonta within the crown for this analysis, based on the poor support values for a stem position found by the analyses positing this relationship (*Krause et al., 2020*; *Celik & Phillips, 2020*) and the findings of *Luo, Kielan-Jaworowska & Cifelli (2002)* which included a broader sampling of species within this lineage and relevant characters than many other analyses. But it should be noted that using a stem position for Euthriconodonta would remove a large number of mid-late Jurassic species from the crown, and potentially shift the crown radiation to later in the Cretaceous.

During the initial stages of the Lower Cretaceous, there is an apparent revival of stem mammal diversity, with both their median and proportionate local richness being similar to or in some datasets exceeding that of crown mammals. However, again, there is a strong indication that this is an artefact of spatial sampling biases. Between the late Berriasian and the Hauterivian, stem mammals are known from only a single locality of uncertain age (the Teete Locality of Russia, Batylykh formation (*Averianov et al., 2018*; *Averianov et al., 2021*)) (Fig. 4D). Within this locality, between 60 and 80% of species are crown mammals, depending on whether haramiyidans are considered crown or stem. All other sampled localities contain only crown mammals. The high apparent richness of stem mammals during this time may therefore represent only the signal of a single, far-north locality to which stem mammals may be geographically restricted. As this locality lay within the Arctic Circle during the Lower Cretaceous, it is possible that the locality represents an extreme refugia where unusual, "relict" faunas are able to survive, as has been posited for other high latitude localities in other time intervals. Alternatively, it may be that the poor geographic sampling during these stages, restricted entirely to the northern continent of Laurasia, is responsible for the lack of stem-mammal-bearing localities. In either event, the apparently high diversity of stem mammals during the Lower Cretaceous should not be treated as a global signal.

Global sampling of synapsids improves during the Barremian, Aptian and Albian, with sampling in both Laurasia and Gondwana, yet stem mammals are still only known from a spatially restricted set of localities(Russia and Japan). Moreover, at this point their species richness, both absolute and relative to crown mammals, decreases. Crown mammal diversity at this point rises to a level that would be maintained for the rest of the Cretaceous (Fig. 1). This radiation of crown mammals has been linked to the ecological re-organisation of terrestrial ecosystems driven by the radiation of angiosperms and associated insect groups

between 100 and 50 million years ago (*Meredith et al., 2011*; *Chen, Strömberg & Wilson, 2019*; *Benton, Wilf & Sauquet, 2022*). During the early stages of angiosperm diversification, mammals (therians and their stem) exhibit low morphological disparity (*Grossnickle & Polly, 2013*; *Wilson et al., 2012*), and it has been noted that several stem lineages disappear during this event (*Luo, 2007*; *Grossnickle & Polly, 2013*; *Benson et al., 2013*). As angiosperms continue to diversify, crown mammal lineages (including therians, multituberculates and dryolestids) increase both their species richness (Figs. 1 and 2) and their morphological diversity (*Grossnickle & Newham, 2016*; *Grossnickle, Smith & Wilson, 2019*).

In fact, if haramiyidans are considered crown mammals, then only one uncertain representative of the mammalian stem (the Australian probainognathian) is known from the Upper Cretaceous; it is possible that the Angiosperm Revolution marked the final replacement of the mammalian stem by the crown. If haramiyidans are considered stem mammals, however, they represent a continuation of the stem not only into the upper Cretaceous, but beyond the end Cretaceous mass extinction, with gondwanatherians (assigned to haramiyidans in this study) persisting into the Eocene (*Goin et al., 2012*). An alternative interpretation, found by *Luo et al. (2015a)* and *Luo et al. (2015b)* but not tested here, is that the majority of Haramyida are stem mammals but Gondwanatherians are within the crown. If this is the case, then the mammalian stem is again inferred to have finally died out around the time of the angiosperm terrestrial revolution. In any case, by the latest Upper Cretaceous, gonwanatherian haramiyidans are not only at considerably lower diversity than crown mammals, but are spatially limited to southern Gondwana (Argentina, Madagascar and India) (Fig. 4E).

## CONCLUSIONS

Unlike the radiation of other modern lineages such as birds, it is difficult to tie the decline and disappearance of the stem lineages of mammals to a particular event like the end Cretaceous mass extinction. In part this is due to the tendency to consider the evolution of the mammalian crown and stem in separation. On analysing the two groups in a single dataset and analytical framework, it is possible to identify a two-phase decline of the mammalian stem and replacement by the crown mammals that persist into the present day. The first phase occurred between the Triassic and Middle Jurassic. Although the lack of substantial sampling in the Lower Jurassic prevents precise tracking of the progress of this phase, by the middle Jurassic the stem mammals were more restricted in their geographic range than the crown mammals, although within localities that they are found their species richness remained at similar levels to that seen in the Triassic. The second phase was a decline in species richness, which occurred during the Lower Cretaceous, although again poor sampling during this interval again prevents precise identification of the timing and duration of this decline. By the Upper Creteaceous, stem mammals were either entirely extinct or restricted to a few lineages in the southern hemisphere (depending on the relationships of Gondwanatherians).

## ACKNOWLEDGEMENTS

I would like to thank Gemma Louise Benevento and Elsa Panciroli for helpful discussion. Suresh Singh, David Grossnickle and an anonymous reviewer also provided useful comments that greatly improved the article. I would also like to thank all those who have uploaded data to the Paleobiology Database, in particular those who made substantial contributions to the dataset used here: John Alroy, Roger Benson, Matt Carrano, Pat Holroyd and Philip Mannion. This publication is Paleobiology Database Official Publication Number 475.

### Funding

The authors received no funding for this work.

### Competing Interests

The authors declare there are no competing interests.

### Author Contributions

- Neil Brocklehurst conceived and designed the experiments, performed the experiments, analyzed the data, prepared figures and/or tables, authored or reviewed drafts of the article, and approved the final draft.

### Data Availability

The data and analytical code from the Paleobiology database is available in the Supplementary Files.

### Supplemental Information

Supplemental information for this article can be found online at http://dx.doi.org/10.7717/peerj.17004#supplemental-information.

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
