# Peer review of "The decline and fall of the mammalian stem"

_PeerJ, doi:10.7717/peerj.17004_

## Round 0.1 · original submission · Minor Revisions

The overall manuscript is well written and the figures are appropriate. I have nothing major to add that has not been already addressed in detail by the reviewers.

·

Basic reporting

The present manuscript is well written, being very clear and generally to the point except for a little elaboration in a couple places (see minor comments for where in text). There are also some brief examples of close repetition, and I think that perhaps creating a dedicated section in the discussion on the sampling issues that impact the study would help the structure and overall succinctness of the manuscript. Nonetheless, this is a clear and direct manuscript that follows the traditional structure of a scientific article as directed by PeerJ.

The figures are appropriate. They are clear, although I think could be improved by having a distinct key in each figure as such details are currently only covered in the figure captions. There are some minor edits that can be made to further improve their readability and link to the text. (See minor comments.)

There is good use of literature, although my knowledge is based on earlier stem mammals rather than the crown so perhaps another reviewer could assess whether this holds for the relevant crown mammals references. In some places, relevant literature is cited but more detail could be given in the present text to help readers better understand the ideas being discussed. (See minor comments for where in text.)

Experimental design

The author makes clear justification for this study, which is a meaningful contribution to synapsid evolutionary biology/palaeobiology. I think the decision to study trends in species richness at a global and local level is well explained and appropriate for this research project, as is the decision to also assess geographic range evolution.
However, I would have liked some more background information for the current hypotheses regarding the changing fortunes of stem and crown mammals through the Mesozoic in the introduction to better set up the scenarios later raised in the discussion. I think there is more focus on describing the overall pattern of the synapsid faunal change through the Mesozoic in the intro and more detail could be given on how the current lack of a single framework for stem and crown mammals has limited our understanding of the rise of crown mammals.

I think detail should be given in the main text of the data that excluded from the original PBDB dataset - how many occurrences and an overview of why (- the specifics are well-handled in the supplement). I also wonder whether the author could run their dataset through the new FossilBrush R package to further ensure that their PBDB data is free of irregularities?
FossilBrush R package reference: Flannery‐Sutherland, J. T., Raja, N. B., Kocsis, Á. T., & Kiessling, W. (2022). fossilbrush: An R package for automated detection and resolution of anomalies in palaeontological occurrence data. Methods in Ecology and Evolution, 13(11), 2404-2418.

Validity of the findings

The findings are interesting and deserve to be published. However, I feel that the analysis of geographic range has issues. The logic of the approach is sound, and I applaud the author for trying to push forward with this to delve deeper into patterns of synapsid evolution through the Mesozoic, but I fear the data simply isn’t sufficient for the broader findings of this part of the study to be robust.
The issue is that the synapsid fossil record and so their occurrence data is especially patchy through intervals of the Mesozoic. Indeed, the author writes at length in the discussion that many of these findings are heavily caveated. It’s good that the author has acknowledged this and is frank about it, and I appreciate that almost all palaeobiological studies must contend with sampling issues, but I wonder whether the scale of these caveats here, means that further analytical study is needed to support the present findings?
I suggest perhaps running a phylogenetic biogeographic analysis using BioGeoBEARS R package to try and incorporate unsampled lineages through the poorly sampled time intervals? Or perhaps include addition reruns of the present analyses focused on particularly well-sampled regions? I understand that this may be beyond the current scope of the study, but I feel that the geographic range part of this study would benefit from further support. Otherwise, I wonder whether it should be saved for a subsequent paper that builds on the species and locality richness results here, in which geographic range changes can be more comprehensively assessed.
BioGeoBEARS R package reference: Matzke, N. J. (2013). Probabilistic historical biogeography: New models for founder-event speciation, imperfect detection, and fossils allow improved accuracy and model-testing. Frontiers of Biogeography, 5(4), 242–248. https://doi.org/10.21425/f5fbg19694

Another potential issue is that Triassic stems mammals such as cynodonts, therocephalians and dicynodonts, could be fairly large and generally possessed quite robust skeletal anatomy, whereas mammaliaformes from the Jurassic onwards are quite small and fragile. This change in the synapsid anatomy and so switch from a macro- to micro-fossil dominated fossil record is presumably going to have an effect on the patterns being investigated. I wonder whether this additional taphonomic bias could be assessed in the present study? Regardless, I think this should be mentioned if not investigated here.

Additional comments

Minor Comments:
1. Abstract. Line 6-7: Perhaps rephrase the latter part of the opening sentence, “The mammalian crown originated during the Mesozoic and radiated rapidly into the substantial array of forms now extant”, as it presently may suggest that all extant mammal disparity emerged rapidly following crown mammal emergence within the Mesozoic.

2. Abstract. Line 10-11: I think the ending of the sentence, “but by the end of the Cretaceous almost all representatives of the mammal line preserved in the fossil record are crown mammals.” is a little wordy and would be much clearer if reworded to something such as, “but only crown mammals are recorded in the fossil record by the end of the Cretaceous”.

3. Abstract. Line 16-17: While I appreciate this is the abstract, this outline of the study seems somewhat vague so I wonder whether you could be a little more specific, by mentioning you used of both species richness and biogeography to study stem mammal decline?

4. Abstract. Line 18-21: I found the following sentence a little confusing at first, “The first phase occurred between the Triassic and Middle Jurassic, during which the stem mammals were more restricted in their geographic range than the crown mammals, although within localities that they are found their species richness remained at similar levels to that seen in the Triassic.”, as you appear to be distinctly comparing the Jurassic with Triassic, but it could be that you’re talking about the span between the Triassic to Middle Jurassic. To avoid confusion, perhaps rephrase to something like, “during which stem mammals were restricted in their geographic range, but their species richness remained constant”?

5. Abstract. Line 24: I think that given the strong impact of poor sampling on your results that are laudably acknowledged in the discussion, some mention of this/caveat of the biogeographic results must be included in the abstract.

6. Introduction. Line 31: Perhaps replace “These stem mammals” with: “Some of these stem mammals”, as currently may imply that stem mammals were a homogenous bunch with all the Palaeozoic diversity persisting past the EPME and ETME. I also think it would be good to include a brief reference here to how ‘stem mammal’ encompasses forms that showcase the transition from more reptilian to mammalian anatomy.

7. Introduction. Line 31: Typo: “well-supported representatives stem” – cut the ‘s’ off of "representatives".

8. Introduction. Line 83-86: “There has been no analysis documenting the diversification, either morphological or species, of Synapsida as a whole throughout the Mesozoic, rendering the transition between stem-dominated and crown-dominate ecosystems unclear.” - Not anymore. The author probably missed the following paper as I think it was published around the time this manuscript was submitted, but I think it should be included here given the focus on diversity/disparity studies spanning stem and crown mammal evolution.
Paper: Hellert, Spencer M., et al. "Derived faunivores are the forerunners of major synapsid radiations." Nature Ecology & Evolution (2023): 1-11.

9. Introduction. Line 93-94: As in the abstract, but perhaps more importantly here, I think some more detail is needed on precisely what sorts of analyses of species richness and distributions were run as this is a little vague - a summary sentence would suffice to highlight the analytical rigour of the study.

10. Materials and Methods. Line 123: Perhaps change “one ensures” to just “ensures” as the former is a little archaic?

11. Materials and Methods. Line 172:173: Perhaps rephrase the latter part of the sentence to “- heterogeneity in geographic and temporal sampling” as I think it’s clearer than the present version?

12. Materials and Methods. Line 178: I appreciate that this method is comprehensively described in the cited Close et al., 2017 paper but I think it would be helpful for readers if you included some more detail of it here. Not much - just a general overview to show that this approach addresses the geographic and temporal sampling issues you outlined in the previous paragraph.

13. Materials and Methods. Line 184: I think it would be good to add in additional emphasis here that these geographic ranges are based on data subject to similar biases and therefore appropriate for comparing patterns across stem and crown mammals - to be clear that this is a comparative study and these results do not show absolute range sizes. I think this framing is important for this aspect of the study.

14. Results. Line 191: How many bioregions are there for each time interval? Wouldn’t the no. of bioregions make a difference to the overall value for that timebin and so trends through time?

15. Results. Line 192-194: The Late Triassic (along with the Middle Jurassic and Early Cretaceous) is generally poorly sampled, meaning these results, while interesting in their own right, cannot be relied upon for temporal trends? Also, you raise the issue of ‘lagerstätte’ bias here, so perhaps you could try to outline what impact these particularly fossiliferous sites have on overall patterns recovered?
Furthermore, could you potentially address these sampling gaps by incorporating phylogenetic methods to include unsampled lineages? Also, a typo here: misspelt synapsids in line 192.

16. Results. Line 192-194: I would suggest cutting “are during the Rhaetian” from this sentence as it sounds a little verbose, and instead start the sentence with “Rhaetian localities”.

17. Results. Line 205-206: Doesn’t this come back to the sampling issue? I think maybe changing to using timebins at stage level, and then switching to epoch or combining stages through these poorly sampled intervals would partially mitigate this issue with poor sampling. It would limit what you can say about these times but what can be said would be more robust.

18. Discussion. Line 251-257: This section seems rather descriptive and largely repeats the geographic range part of the results section, so perhaps cut this and/merge into that section of the results?

19. Discussion. Line 259: I don’t think it’s certain that this pattern is true, given the sampling issues so be sure to temper such statements with caution.

20. Discussion. Line 270: Is competitive replacement by crown mammals a strong contender for the decline of stem synapsids? Here, I think there should be some more detail regarding this concept and its possibility in this scenario.

21. Discussion. Line 292-294: Maybe rerun the geographic analyses but on a smaller scale focusing on particularly well-sampled regions during different intervals, as an additional point of comparison to the overall patterns?

22. Discussion. Line 319: Typo here: missing word following morphological – either diversification or radiation?

23. Discussion. Line 322-324: Perhaps cut the first part of this opening sentence up to the point, “if we discount -” as it’s unnecessary repetition.

24. Conclusion. Line 334: I would also cut the opening part of this opening sentence up to the point, “it is difficult -” as it also closely repeats this point from the intro. Also, I would then rephrase to, “It has been difficult -”.

25. Figures 1-3: I think it would be helpful for readers if you could include shaded bars to highlight the phases of decline outlined in the text, but this may be a personal preference so I leave it to the author to decide.

26. Figure 4: Please increase the point size in this figure as the different points are hard to distinguish at present.

·

Basic reporting

Your study is fairly straightforward, tackling a simple but interesting question: when did crown mammals replace stem mammals as the dominant taxa in mammalian faunas? The data is from PBDB, which has its many faults, but the analyses seem robust and I doubt that cleaning up the PBDB occurrence data would significantly alter your results/conclusions. I don’t think that the resulting diversity patterns will surprise anyone who works on these taxa. However, like you mention, researchers usually focus on cynodont-only or mammal-only datasets, and thus your analyses are novel in that they more broadly examine synapsid patterns compared to previous studies. And you apply diversity analysis methods that haven’t previously been applied specifically to a synapsid dataset, at least as far as I’m aware. Thus, I think that your study is a worthwhile contribution that improves our understanding of synapsid evolution.
I have two major suggestions/concerns, which I explain below. In sum, I think that the study would more strongly contribute to our understanding of the origins of modern biodiversity if you examined therians vs non-therians (or cladotherians vs non-cladotherians) rather than stem vs crown mammals. However, that’s more of a subjective opinion rather than a critical issue, so I’ll leave it up to you to decide whether to follow that suggestion.

Feel free to contact me if you have any questions on my review.

Sincerely,
Dave Grossnickle
david.grossnickle@oit.edu


My two major concerns/suggestions related to your decision to examine stem vs crown mammals:

1. Why stem vs crown? Is the crown node just an arbitrary cutoff? Or are there other biological, phylogenetic, etc reasons behind that decision? It often makes sense to focus on crown vs stem, especially because of our biased interest in modern/crown origins. But for mammals, monotremes are super weird and much more similar to late ‘mammaliaforms’ than to other extant mammals (therians). So, by using crown mammals, you end up lumping a lot of weird early mammals (e.g. eutriconodonts, multituberculates) in with the more derived, ecologically diverse therians, and it may muddle the story.

I think a more interesting comparison would be therians vs non-therians, as therians represent the ‘true’ radiation of modern mammals. It'd be a better test of when modern mammal groups replaced earlier groups, helping inform our understanding of the origins of modern biodiversity (e.g. see Hellert et al. 2023 Nature E&E for an example of examining therians vs non-therian mammaliaforms). Further, a therian vs non-therian analysis would complement your recent disparity paper (Brocklehurst et al. 2021 Curr Biol) that includes therian-only analyses.

A possible problem is that therians don’t really diversify taxonomically until the Early Cretaceous, so your Triassic-Jurassic results are going to be very boring, i.e. just non-therians. Also, multituberculates (non-therians) are still abundant in the Paleocene, so you might want to include Paleocene analyses if you follow my suggestion. That complicates things. But I still think that examining therians would be more informative in terms of understanding the origins of modern biodiversity, which makes for a more interesting evolutionary story.

Or instead of therians, you could make an argument for examining cladotherians (therians and close relatives) to represent the origins of modern biodiversity – the cladotherian node likely represents key shifts in molar morphology, jaw morphology, chewing function, and maybe middle ear-jaw separation (Patterson 1956 Fieldiana, Grossnickle 2017 Sci Rep, Bhullar et al. 2019 Nature, Grossnickle et al. 2021 ZJLS). (In my studies I’ve argued it’s a more important node than the therian node.) And the S. American diversification of early cladotherians (dryolestoids) seemed to mimic the Laurasian diversification of therians. Including cladotherians would give you a sample of Late Jurassic taxa.

If you disagree with my suggestion, that’s fine, but I simply recommend that you defend (in your paper) your choice to compare stem vs crown mammals rather than other options (e.g. therians vs non-therians, or cladotherians vs non-cladotherians).

2. There’s an enormous amount of topological uncertainty in the early mammal tree, meaning that you can’t easily differentiate crown taxa from stem taxa. The most contentious debate is on haramiyidans (misspelled as “haramyidans” in the paper), and I like that you re-ran analyses using 3 possible positions for haramiyidans. That helps ease some concerns. However, there’s also uncertainty on the relationships of other mammal/mammaliaform groups. For instance, eutriconodontans are often recovered as stem mammals (e.g. Rougier et a. 2011 Nature, 2012 PNAS, Krause et al. 2020 Nature). Further, Tom Rich and colleagues argue that australosphenidans are closely related to therians (i.e. they share homologous tribosphenic molars). I don’t think that he’s run phylogenetic analyses, but australosphenidans are often considered the earliest branching mammals (i.e. stem monotremes), and moving them in the tree toward Theria would likely kick out other early mammal groups (maybe multituberculates) from the crown. Also, if multituberculates are descendants of early haramiyidans (Jin Meng’s argument) and early haramiyidans are stem (Zhe-Xi Luo’s argument), then it’s very possible that haramiyidans+multituberculates are stem mammals. Removing multituberculates from the crown would certainly have a big influence on your results. Also, there are many unstable singleton taxa ‘floating’ around the Mammalia node, including Tinodon, Kuehneotherium, and Fruitafossor. Anyway, my point is that it’s very uncertain as to which taxa are crown and which are stem.

You’d largely alleviate the phylogenetic uncertainty issue if you use therians vs non-therians, like I suggest above. There’s a little uncertainty about whether the earliest therians (e.g. Juramaia) are crown or stem therians, but those issues are relatively minor compared to the issues with Mammalia.

And, again, if you decide not to follow my advice, then, at minimum, I recommend adding more justification for your decisions to include some taxa as stem or crown. For example, you should explain your choice to include Eutriconodonta in the crown or stem, and cite relevant sources.

Line comments

Line 13: “bot” should be “both”

Line 22: You switch between using “Lower Cretaceous/Jurassic” and “Early Cretaceous/Jurassic” throughout the manuscript. My tendency is to use Early because I’m usually referring to time periods, whereas I think of Lower as referring to rock layers. But maybe that’s just personal preference or maybe you have a reason for switching between the two.

Line 36: I recommend clarifying that the 3 listed groups are just the extant representatives of crown Mammalia – there are many other extinct groups of crown mammals.

Line 38-54: You could note in this paragraph that the inclusion or exclusion of early haramiyidans within crown mammals influences the estimated age of the mammalian node by ~40 million years (see the Davis & Cifelli 2013 Nature commentary paper, and Luo et al. 2015 PNAS Fig 4).

Line 46 (and elsewhere): I believe “haramyidans” should be “haramiyidans” (added “i”), and “Haramyavidae” (Line 51) should be “Haramiyaviidae” (added two “i”s). It looks like these misspellings are constant throughout the paper.

Line 50: “King & Beck 2022” should be “King & Beck 2020”. Same for Line 112. Also, I think Krause et al. (2020 Nature) recovered a similar polyphyletic Haramiyida. (Maybe also recovered by Hoffman et al. 2020 – the phylogeny paper from the JVP memoir on Adalatherium).

Line 61: What about Chronoperates paradoxus, the cynodont from the Paleocene (Fox et al 1992 Nature)? Or has that been reclassified? PBDB has it listed as a cynodont.

Line 73 (and elsewhere): Rather than “gondwanatherian haramyidans”, I recommend just “gondwanatherians”. Traditionally, Allotheria included Gondwanatheria, Haramiyida, and Multituberculata, with lots of uncertainty and debate over the years as to how those 3 groups are related. But I don’t think gondwanatherians are usually classified as a type of haramiyidan, although I realize they’re often recovered in phylogenies as within Haramiyida (e.g. Huttenlocker et al. 2018, King & Beck 2020). But multituberculates are also often recovered as being within Haramiyida (e.g. Jin Meng papers) and they aren’t called “multituberculate haramiyidans.”

Line 102-104: It’s fine to use PBDB taxonomy, but that taxonomy is based on phylogenetic analyses, which are a mess (see my comments above). For example, if PBDB currently classifies eutriconodontans as crown, then whoever last entered/updated that classification info was relying on a specific phylogeny that recovered eutriconodonts as crown (e.g. Zhe-Xi Luo or Jin Meng phylogenies).
I spent days if not weeks trying to clean up PBDB taxonomy for our synapsid metatree in Hellert et al. 2023, so I updated a lot of the early mammal taxonomy. I generally followed Luo’s phylogenies, but I also considered results from Meng, Rougier, Krause, King & Beck, etc. It might be worth citing some of these papers to help justify why some taxa were included within (or excluded from) crown Mammalia.

Line 109: Based on “Gondwanatherian haramyidans” (Line 73), I assume you included gondwanatherians with haramiyidans? I recommend explicitly stating this because not all researchers would agree that gondwanatherians are haramiyidans. And I don’t think that gondwanatherians are classified within Haramiyida in PBDB.

Line 176: “synapsuids” should be “synapsids”

Line 243: “lower” should probably be capitalized.

Line 269-277: This is interesting. Do you think this ties into the observation that extinction often precedes diversification and/or there’s ‘survival of the novel’ (e.g. Brocklehurst 2015, Hellert et al. 2023)? Or are there other macroevolutionary patterns at play? In any case, you might expand on this finding a bit because it might be of interest to readers.

Line 288: I’m a little confused here. By “morphological diversification of eutherian mammals” do you mean the Late Cretaceous/Paleocene diversification of eutherians? And why “eutherians” and not “therians”? (There’s only one eutherian/therian from the Jurassic, Juramaia, and its geologic age and phylogenetic position are questionable, so I wouldn’t infer too much from that single taxon.) I recommend adding more details for clarity.

Line 316: I recommend adding a rough age range for the KTR. Also, note that Benton suggested replacing the KTR with the Angiosperm Terrestrial Radiation (I think he defines it as ~100-50 Ma). The ATR probably better aligns with the rise of therians and multituberculates in the latest Cretaceous (see comment below).

Line 319: “a morphological of” should maybe be “a morphological diversification of”?

Line 318-320: Kind of, but I think the details are a little jumbled here. I argued that the KTR (~125-80 Ma) was associated with a decrease in disparity and high turnover of clades (Grossnickle & Polly 2013), also shown by very low disparity in therian/multituberculate molars during the KTR (Wilson et al. 2012, Grossnickle & Newham 2016). And then in the wake of the KTR, starting ~80 Ma, there were diversifications of therians (Grossnickle & Newham 2016, Grossnickle et al. 2019), multituberculates (Wilson et al. 2012 Nature) and dryolestoids in S. America (see sources in Grossnickle et al. 2019). I wouldn’t really associate these latest Cretaceous diversifications with the KTR because they occur post-KTR. They’re more likely associated with the ecological diversification of angiosperms in the latest Cretaceous (e.g. see the leaf vein density in Wilson et al. 2012). It might be more appropriate to link these mammal diversifications with the ATR, not the KTR. During the mid-Cretaceous (middle of KTR) the angiosperms diversified taxonomically, but not ecologically, i.e. they were mostly small ‘weeds’. It wasn’t until the latest Cretaceous when the ecological diversification of angiosperms took off (e.g. increased seed sizes, leaf vein density, woody plants, etc). Anyway, sorry for the longwinded comment! I just recommend revising your sentences a little to clarify that the ecological diversification of mammal groups probably wasn’t linked to the KTR.

Line 326: Again, there might be a Paleocene cynodont (Fox et al 1992), but I don’t know how well accepted that is.

Line 326: Are you referring to gondwanatherians here? As noted above, not everyone considers them to be haramiyidans, and even the researchers working on them (e.g. Krause, Hoffman) are highly uncertain about their phylogenetic placement. So I don’t recommend calling them haramiyidans, or, if you do, make it clear early in the paper that you’re classifying them here as haramiyidans.

Line 334-336: As I explain above, I think a therian vs non-therian analysis would be more informative. The timing and pattern of the therian diversification would more closely match that of birds, although it probably wouldn’t be as closely linked to the K-Pg (and maybe instead linked to the ATR or Paleocene events, e.g. Brocklehurst et al. 2021). If not for a couple of stubborn monotreme lineages, therians would be the crown mammals. They represent almost all extant mammal diversity. I think focusing on them would be a more interesting story.

Figure 4: I can’t see some of the points in the figure, especially the haramiyidans (gray on gray background), although maybe that’s because PeerJ’s file is low-res. In any case, you may want to increase the size of the points.

Experimental design

no comment

Validity of the findings

no comment

Cite this review as

Reviewer 3 ·

Basic reporting

.

Experimental design

.

Validity of the findings

.

Additional comments

This manuscript is a useful research with insight. I recommend that it can be accepted for publication by PEERJ after revision and improvement on reference citation.

Starting from the earliest-known cynodonts in the latest Permian, the cynodont-mammal evolution shows successive episodes or diversification: the stem mammaliamorphs (cynognathians, probaingnathians, tritylodontids, and tritheledontids), the stem mammaliaform clades, and the fossil clades nested within the crown mammals are all distinct groups phylogenetically. Each of these mammaliamorph and mammaliaform clades is a cluster of taxa that diversified in their own right, had their evolutionary trajectories, and finally ended in extinction without “giving rise to” known descendants. Dr. Brocklehurst is seeking to tease apart the evolutionary trajectories of stem mammaliamorph cynodonts (all extinct by after Aptian-Albian), and of the stem mammaliaforms (all extinct by K-Pg boundary), from the trajectories of clades nested in the crown Mammalia. The key observation is that the decline of stem mammaliamorphs and mammaliaforms had two episodes (Mid-Late Jurassic, and then Early Cretaceous). Both phases are relatively long, and un-related to major geological event or abrupt faunal turnover. I agree with the observation and the conclusions.

Perhaps also interesting are the methods of the author. In addition to the typical summary of “species richness” (standing taxonomic diversity), the author also used proportion of “stem-mammaliamorphs” to the crown Mammalian diversity, and the diversity after correction for size of the geographic area. All three sets of analyses are confirming a similar pattern. This is nice.

Fossil mammaliamorphs (also known as “Prozostrodonts” by some authors, including brasilidontids; tritylodontans, and tritheledontids) did not survive after Aptian-Albian of Early Cretaceous (no late Cretaceous members are known). Fossil mammaliaforms (Sinoconodon, morganucodontans, kuehneotheriids, docodontans, and haramiyidans) had decent diversity and a Pangean distribution (morganucodonts) in Early Jurassic, and peaked in diversity in Mid-Late Jurassic (mostly docodonts and some haramiyidans), but declined in Early Cretaceous. Gondwanatherians, which are a subclade of larger haramiyidan clade of the mammal-stem (or close relatives to multituberculates) also perished by Cretaceous end, if Greniodon (Goin et al. 2012) can be ruled out as a gondwanatherian (see Rougier et al. 2021 discussion on this topic).

Revision Requirement

However, this paper is deficient in proper vetting on references, and in reference citations. This is a major lapse that must be corrected. Also, there are a couple typos.

Line 13 -14. Analyses of “bot” species macroevolutionary patterns tend to focus on either the Mammaliaformes or the non-mammalian cynodonts. “bot” is a typo.

Lines 23-24. Re “results show the decline of stem mammals…” Can I suggest this be changed to make it more explicit: “results show the decline of stem mammals, such as tritylodonts and several mammaliaform groups…”

Lines 34-35. Re: “ The identity and age of the youngest stem mammal is uncertain, but they potentially survived into the earliest Cenozoic (Huttenlocker et al 2018)”

This sentence is hinting on the taxonomic uncertainty of Greniodon (by Goin et al. 2012). To make it more direct and unambgiuous, I suggest that the author should cite several papers together (Goin et al. 2012; Huttenlocker et al. 2018; Rougier et al. 2021) here; these papers together outlined the uncertainty with the interpretation of Greniodon, but more clearly explained by Rougier et al. (2021).

Lines 38-43. von “Heune” (1940) is a typo. Should be von “Huene”. The author has shown a major lapse on his understanding of the taxonomic literature here. Either the author is not familiar with the history of systematic paleontology haramiyidans, or that the Paleobiology Database (PBDB) had limited information, while Dr. Brocklehurst relied on PBDB’s incomplete information to make his own comments in Lines 38-41. In short, the Late Triassic mammaliaform fossils from St-Nicolas-de-Port site mentioned first by Sigogneau-Russell (1983), have been re-studied by herself (Sigogneau-Russell et al. 1986). And then revised and augmented with new discovery. Von Huene (1940) is totally out of date. Here is my explanation:

Sigogneau-Russell (1983), which was cited by Brocklehurst in this location, did not attempt to establish the formal taxonomy of the three specimens found in the Rhaetian St-Nicolas-de-Port (SNP) site in France. She only tentatively referred these to ?”Multituberberculata”. However, not long after, these same specimens were re-assigned to a new genus (Theroteinus), in its own unique family Theroteinidae (Haramiyida) by Sigogneau-Russell herself (Sigogneau-Russell et al. 1986). Hahn et al. (1989) further elevated the Theroteinidae family to the new order Therotainida.

The specimens SNP61 and SNP 78 (the lower and upper molars respectively) in Sigogneau-Russell’s 1983 paper, later became the type and referred specimens of Theroteinus nikolai; SNP 2 was now an upper molar of Theroteinus rosieriensis (Theroteinidae, Haramiyida). These species are now independently vetted by Kielan-Jaworowska et al. (2004) book, and most recently re-studied in details with full CT scanning by Maxime Dabuysshere (2016).

Gerhard Hahn (1987) mentioned a questionable multituberculate - Mojo usuratus - from Habay-la-Vieille II site (Gaum, Belgium). But this “species” was based on half of an upper premolar and its incomplete crown already lost all features to wearing (Kielan-Jaworowska et al. 2004: fig. 8.31 on page 311). Kielan-Jaworowska re-examined this fossil and argued that it is highly unlikely that Mojo would be any multituberculate at all (Kielan-Jaworowska et al. 2004: Table 2.1; page 310; fig. 8.31 on P. 311).

Thus the St-Nicolas-de-Port site and the Habay-la-Vieille II site have no multituberculates. These two key sites in Europe of Rhaetian age have no fossils that can be assigned to the crown Mammalia.

I suspect that the author just used PBDB’s easy-to-access information for this discussion(Lines 38-4) , without awareness PBDB’s information can be incomplete or wrong, and must be verified, or the basic systematic paleontology literature is just beyond the author’s bandwidth to deal with. Whatever the reason, it is not right.

Lines 49-53. King and Beck “(2022)” has a typo. Should be (2020). Several key references should be added here. Zhou et al (2013) was the first paper using large scale phylogenetic dataset to show Haramiyidae and Haramyavidae are on mammal stem, unrelated to multituberculates and excluded from the crown Mammalia. Luo et al (2015: fig 4) show how much a time difference (circa 30 mya) between the basal placement of Haramiyida (Zhou et al. 2013; Luo et al. 2015; Huttenlocker et al. 2018) and a placement of Haramiyavia in the crown Mammalia (Bi et al. 2014).

Krause et al. (2020, in Nature) placed Haramiyidae and Haramiyavidae on the mammal stem, but kept Gondwanatherians and multituberculates in the crown, on a cladistic matrix developed by themselves. The Krause 2020 work is independent from analysis of King and Beck (2020), which was based on a simplified version of the Huttenlocker et al. 2018 dataset. With these explanation, I suggest the author revised the paragraph (Lines 39-53) to

“However, other authors found haramyidans to be stem mammals Zhou et al. 2013; Luo et al. 2015; Huttenlocker et al. 2018). Still other analyses, such as Krause et al. (2020) and King and Beck (2020) who used different phylogenetic datasets, found a polyphyletic Haramyida, with Haramyidae and Haramyavidae (the lineages known from the Triassic) excluded from the mammalian crown. These two alternative placements of Haramyidae and Haramyavidae would result in two different age estimates of crown Mammalia, by about 25-30 mya (see Luo et al. 2015, fig 4). Nevertheless, despite this uncertainty surrounding the origin of modern mammals, a Late Triassic-Early Jurassic origin of crown mammals is consistent with estimates derived from molecular clocks (Meredith et al. 2011; dos Reis et al 2012, 2015).”

Also I would suggest that the author check Upham et al. (2019) Plos-Biology. These authors used the node-dated fossilized birth-death Bayesian model to estimate the origins of crown Mammalia to be from Late Jurassic (Norian) to Middle Jurassic (Upham et al. 2019: Fig. 2c).

Lines 73-74. The question remains to be open whether Greniodon (Goin et al. 2012) is a ferugliotherid or a sudamericid within the gondwanatherian. This uncertainty opens to alternative interpretation. Rougier et al. (2021: p. 308-309) accepted the Goin (2012) interpretation that Greniodon is gondwanatherian, thus extending this clade into Eocene. Rougier et al. (2021) should be cited here.

Lines 108-112. I urge the author add key refs (red font) in this paragraph. Also a correction on typo of King and Beck (2020):

From this dataset, three test datasets were created, with different assignments for Haramyida as discussed above. The first assigned Haramyida to the mammalian crown (Bi et al. 2014); the second the mammalian stem (Luo et al. 2015); the third treated Haramyidae and Haramyaviidae as stem mammals, and separated from multituberculates that are undoubtedly crown mammals, a set of relationships found independently by Krause et al. (2020) and King & Beck (2020)”

Lines 192-196. For the high diversity of the named fossil taxa from the Habay-la-Vieille II site, the author should really cite Pascal Godefroit (1999) original study of this site.

Line 279. “pioscvores” is a typo. Should be “piscivores”

References cited in this review and should be added to the manuscript during the revision:

Debuysschere, Maxime. 2016: A reappraisal of Theroteinus (Haramiyida, Mammaliaformes) from the Upper Triassic of Saint-Nicolas-de-Port (France). PeerJ. 2016; 4: e2592.

Godefroit, Pascual (1999). New traversodontid (Therapsida: Cynodontia) teeth from the Upper Triassic of Habay-la-Vieille (southern Belgium). Palaontologische Zeitschrift 73 (3/4): 385-394.

Hahn, G. 1987. Ein Multituberculaten–Zahn aus der Ober-Trias von Gaume (S-Belgien). Bulletin de la Société belge de Géologie 96: 39–47.

Hahn, G., Sigogneau-Russell, D., and Wouters, G. 1989. New data on Theroteinidae—their relations with Paulchoffatiidae and Haramiyidae. Geologica et Paleontologica 23: 205–215.

Kielan-Jaworowska, Z., R. L. Cifelli, and Z.-X. Luo. 2004. Mammals from the Age of Dinosaurs: Origins, Evolution, and Structure. Columbia University Press, New York. Pp. i-xv, 1-630; 239 figures.Krause DW, Hoffmann S, Hu Y, Wible JR, Rougier GW, Kirk EC, Groenke, JR, Rogers RR, Rossie JB, Schultz JA, Evans AR, von Koenigswald W., and LJ Rahantarisoa. 2020. Skeleton of a Cretaceous mammal from Madagascar reflects long-term insularity. Nature 581 (7809), 421-427

Luo, Z.-X., Gatesy, S. M., Jenkins, F. A., Amaral W. W. and N. H. Shubin. 2015. Mandibular and dental characteristics of Late Triassic mammaliaform Haramiyavia and their ramifications for basal mammal evolution. Proceedings of National Academy of Sciences USA. 112 (51): E7101–E7109 (doi: 10.1073/pnas.1519387112).

Rougier, G. W., A. G. Martinelli, and A. M. Forasiepi. 2021. Mesozoic Mammals from South America and Their Forerunners. Springer Earth System Sciences, Switzerland. Pp.1-388.

Upham N. S., Esselstyn, J.A., Jetz W. (2019) Inferring the mammal tree: Species-level sets of phylogenies for questions in ecology, evolution, and conservation. PLoS Biology 17(12): e3000494. (https://doi.org/10.1371/journal.pbio.3000494).

Sigogneau-Russell, D., Frank, R. M., and Hemmerlé, J. 1986. A new family of mammals from the lower part of the French Rhaetic. In K. Padian (ed.), The Beginning of the Age of Dinosaurs, 99–108. Cambridge University Press, Cambridge.

Zhou, C.-F., Wu, S., Martin, T. and Z.-X. Luo. 2013. A Jurassic mammaliaform and the earliest mammalian evolutionary adaptations. Nature. 500: 163-168 (doi:10.1038/nature12429).

Cite this review as

---

## Round 0.2 · Minor Revisions

The overall experimental design and validity of findings are sound. However, please address the comments from Reviewer 3 (and previously Reviewer 2 as well), specifically pertaining to the Luo et al. 2015a PNAS paper, inlcuding figure 4 of this paper; this must be better addressed and corrected before full acceptance.

·

Basic reporting

The manuscript has been improved following the revisions suggested by the other reviewers and me. The writing is clear and professional, and well structured. The rationale and methods are better explained. The few typos and overly verbose passages in the first draft have been removed, although there are a couple typos in this draft:

1. Line 93: Typo: singular rather than plural - "a series of discrete radiation".

2. Line 234: Typo: “syanpsids”.

3. Line 327: Typo: “Triasic”.

4. Lines 329-330: Repetition: “increased sampling of micro-vertebrate sampling”.

I also have an additional comment regarding lines 117-120: I previously suggested including some additional detail on the sorts of analyses of species richness and distributions featured in this study. I appreciate the author’s response, but I feel they may have misunderstood my comment – I was not asking for extensive detail on the methods in the introduction – such detail rightly belongs in the methods. My concern is that this section doesn’t do the methodology justice - while it may not appear so from my queries about sampling, I like their approach and believe it deserves more 'fanfare'. Including just a little more detail of this approach in the intro would better highlight its novelty to readers and may encourage their interest. At a minimum, mention that the study uses an approach based on bioregions, which I do not believe is particularly common in such palaeobiological studies. However, this isn’t a critical revision, so the author is welcome to leave this section as is.

Experimental design

I understand the author’s responses to my previous comments on sampling issues, and accept their reasons for declining my suggestions. These choices do not harm the manuscript as my previous suggestions were aimed at attempting to bolster the present findings of one section with additional, alternative methods rather than addressing critical issues. I appreciate the author's rationale for declining further analyses as being too heavy and broad an endeavour for inclusion in this study. I believe the present manuscript sets up many interesting ideas for future biogeographical exploration of synapsid macroevolution and should be published once the other minor revisions are made.

Validity of the findings

No Comment.

·

Basic reporting

Thanks for addressing all my concerns. I don’t have any further issues to raise, but I responded to a couple of your comments below for your consideration.

“There are two reasons why I placed the division as crown and stem. ...”

Response: Those reasons are fine. As I previously stated, it’s my opinion that a therian vs non-therian comparison would be more interesting and impactful, but you're free to ignore that suggestion. And I meant to mention in my previous review that you could simply add a secondary or supplemental therian vs non-therian analysis – it’d be fairly easy, wouldn’t take away from the central focus on stem vs crown, and would give you more to discuss and likely increase the impact of the study. Something to consider.

“If I understand the papers of Rich and his colleagues correctly (e.g. Rich et al. 2002, J Vert Palaeontol; Flannery et al. 2022, Alcheringa), they are suggesting that austalosphenidans are not stem monotremes, not that they and monotremes are more closely related to therians than, for example, mutlituberculates.”

Response: I agree and didn’t mean to imply otherwise. And I wasn’t suggesting that you should re-run analyses based on this issue. But if Rich and colleagues (which includes other experts in the field such as Jim Hopson and John Flynn) are correct that there’s no australosphenidan + monotreme link, it means we basically have no idea how monotremes relate to early mammal groups. Meaning we have no idea which early mammal groups are crown and which are stem. And, based on some of the scorings in characters matrices (e.g. monotremes lack an ossified meckel’s cartilage like therians), my guess is that by removing the australosphenidan+monotreme connection, monotremes would shift closer to therians in the tree, pushing other groups out of the crown. That’s all very speculative, but I mentioned it in my last review to help illustrate my underlying point – we really have no idea where the crown node is. Every early mammal researcher seems to have a different opinion on it. Which then makes me a little skeptical of any stem vs crown results. My recommendation is to add more clarification in your Discussion that your results are dependent on the phylogenetic hypotheses that you’ve chosen to follow.

Dave Grossnickle
david.grossnickle@oit.edu

Experimental design

No comment

Validity of the findings

No comment

Cite this review as

Reviewer 3 ·

Basic reporting

2nd Review of PEERJ Manuscript “The Decline and Fall of the Mammalian Stem” by Neil Brocklehurst

Referee #3

Again, I support for this paper to be published by PEERJ. There is an inherent value in to contrast and compare the vicissitude of fossils by phylogenetic partition: crown mammal clades vs nonmammalian mammaliaforms.

However, after looking though the explanation of revision, and the revised manuscript, I again spotted an un-intentional errors, which are still with the revised version. I don’t think the author was against the best practice to seeking to understand accurately the relevant literature and then cite it accurately, and in nuanced way if the issues are complex. But I must say that the author had a difficulty to grasp some of the key nuances of the literature, and the author got mixed up on previous papers.

Here I point out that Dr. Brocklehurst did not seem to have paid attention the suggestion from two referees (referee 2 and referee 3), for him to check Luo et al. 2015 PNAS paper, and more specifically to look into figure 4 of this paper. In the revision, Dr. Brocklehurst overlooked this again. This must be corrected before this paper can be accepted.

Luo et al. 2015a (Luo, Gatesy, Jenkins, Amaral and Shubin) in PNAS (doi.org/10.1073/pnas.1519387112) on restudy of Haramiyavia: This is the main paper that highlighted the difference of Haramiyavia-stem vs Haramiyavia-“Crown.” (Luo et al. 2015a: fig 4) – This paper should be cited in the Brocklehurst text in the paragraph of lines 77-90.

However,

Luo et al. 2015b (Luo, Meng, Ji, Liu, Zhang and Neander) in Science (doi.org/10.1126/science.1260880) is a separate study. Luo et al. 2015b is on Docofossor, which is a specialised burrower with hypertrophied autopods and limb features that expanded the ecomorphological disparity of stem mammaliaforms. This paper is relevant to content on lines 413-414.

Please note - There are two different papers, relevant to the content of Brocklehurst manuscript here, in different context! Both should be cited, each for different reason. Dr. Neil Brocklehurst got these mixed up – honestly it is chagrin for this referee to point this out, twice.

It’s really a minor error, but glaring one, because the emphasis of this Brocklehurst work in to tease apart the compound patterns of diversity between the stem mammaliaforms and those of crown mammals, which fundamentally dependent on key interpretation of the stem mammaliaforms.

Luo, Z.X., Gatesy, S.M., Jenkins, F.A., Amaral, W.W. and Shubin, N.H. 2015A. Mandibular and dental characteristics of Late Triassic mammaliaform Haramiyavia and their ramifications for basal mammal evolution.Proceedings of the National Academy of Sciences USA 112(51): E7101-E7109. (doi.org/10.1073/pnas.1519387112)

Luo, Z.-X., Q.-J. Meng, Q. Ji, D. Liu, Y.-G. Zhang, and A. I. Neander. 2015B. Evolutionary development in basal mammaliaforms as revealed by a docodontan. Science 347: 760-764 (doi.org/10.1126/science.1260880)

Experimental design

Not applicable

Validity of the findings

By and large the findings are useful. But it would be better that the author improves on the author's understanding of the key references.

Cite this review as

---

## Round 0.3 · accepted · Accept

The author has assessed and corrected major comments by the reviewers, and the manuscript appears near-ready for publication.